# The Influence of Grassland Ecological Compensation Policy on Grassland Quality: Evidence from the Perspective of Grassland Ecosystem Vulnerability

**Mengmeng Liu [1,2], Wanqing Wu [1,2] and Hua Li [1,2,*]**

1   College of Economics and Management, Northwest A&F University, Xianyang 712100, China; lmm230017@nwafu.edu.cn (M.L.); wwq0096@nwafu.edu.cn (W.W.)
2   Center for Resource Economics and Environment Management, Northwest A&F University, Xianyang 712100, China
*   Correspondence: lihua7485@nwafu.edu.cn

**Abstract:** The grassland ecological compensation policy (GECP) is currently the largest grassland ecosystem payment program in the world, trying to manage and restore degraded grasslands to achieve a virtuous cycle of grassland ecosystems. However, responses to the policy may vary across different regions. Based on panel data from 395 counties in six provinces in China's pastoral areas from 2001 to 2021, this paper uses the difference-in-difference (DID) and moderation effect models to test the relationship between GECP and grassland quality from the perspective of grassland ecosystem vulnerability. The study found the following: (1) The spatial differentiation pattern of grassland ecosystem vulnerability in the six provinces of China's pastoral areas is obvious, and the vulnerability is mainly slight and moderate (2) The GECP generally has a significant positive impact on grassland quality. (3) Grassland ecosystem vulnerability has a negative regulatory effect on the impact of the GECP on grassland quality. The positive impacts of policies become more pronounced as the vulnerability of grassland ecosystems decreases.

**Keywords:** ecological compensation; grassland quality; grassland ecosystem vulnerability

## 1. Introduction

As the world's largest payment for ecosystem services (PES) for grassland ecosystem services [1,2], the Grassland Ecological Compensation Policy (GECP) aims to encourage and supervise herders in reducing livestock pressure by ways of grazing ban and grass–animal balance, so as to alleviate the pressure on the grassland and ultimately restore the grassland ecological environment [3]. Since the implementation of the policy, although the grassland ecological environment in some areas has improved, the overall situation is not optimal [4]. According to the report of the 2021 Forestry and Grassland Protection Conference, nearly 70% of China's grasslands will still be degraded to varying degrees by 2021 (https://www.gov.cn/xinwen/2021-08/20/content_5632429.htm, accessed on 8 August 2023). Furthermore, during the 2021 China Grassland Protection and Restoration Promotion Work Conference, it was indicated that the average livestock overloading rate in China's key natural grasslands for the year 2020 was still 10.1%, despite a decline compared to previous rates (https://www.gov.cn/xinwen/2021-07/19/content_5625864.htm, accessed on 8 August 2023). Additionally, due to herders evading inspections for supplementary incentives, the actual rate of livestock overloading may exceed official statistics. It can be seen that after the implementation of the GECP, the situation of grassland degradation is still severe. So, a scientific evaluation of the real effects of the GECP on improving the ecological compensation mechanism and shaping the next version of the policy is very important from both a theoretical and a practical standpoint.

The Grassland Ecological Compensation Policy (GECP) was formally proposed and implemented by the Chinese government in 2011. This policy divides the grassland in the target area into a grazing ban area and a grass–animal balance area. The grazing ban area is aimed at places where the grassland is seriously degraded, and grazing is completely prohibited according to policy requirements; the grass–animal balance area is for raising a specific number of livestock animals according to the carrying capacity of the grassland. If herders can abide by the policy, they can obtain government subsidies. In essence, the policy aims to guide and encourage herders to coordinate the balance between forage supply capacity and livestock quantity, so as to establish a sustainable grassland utilization mechanism. As of 2022, the GECP has been implemented in three cycles, with each cycle spanning five years. The implementation area has been gradually expanding, starting from the initial coverage of eight grasslands in resource-rich provinces, including Inner Mongolia, Tibet, Ningxia, Qinghai, Sichuan, Xinjiang, Gansu, and Yunnan. Currently, it encompasses 13 provinces. Moreover, the investment funds for the program have remained consistent without any interruptions (https://www.forestry.gov.cn/search/349677, accessed on 8 August 2023).

Will this ambitious payment for ecosystem services program be as effective as expected in reducing the number of grazing livestock animals and curbing grassland degradation? Numerous scholars have engaged in extensive theoretical discussions and empirical testing. Regrettably, a consensus on the current research findings has not been reached [1,5,6]. Certain perspectives posit that stringent environmental regulations and well-targeted financial subsidies can assist herders in undertaking production transformation, leading to a reduction in livestock numbers and facilitating grassland ecological protection [5,6], especially in grass–animal balance areas where controlled grazing is permitted [7]. For example, Liu et al. found that the GECP can effectively reduce the number of sheep and significantly improve the grassland condition with data from 54 counties in the pastoral areas of Inner Mongolia [8]. Hou et al. also discovered that under the GECP, a 1% increase in medium payment intensity corresponds to a 0.011% increase in grassland Normalized Difference Vegetation Index (NDVI) [2]. In contrast, another perspective suggests that the compensation standard in GECP is too low to adequately support herders in completing production transformation [9]. As a result, the policy's effectiveness is limited, especially when policy supervision is relatively lax [10]. In such cases, some herders may misuse supplementary reward funds to expand grazing activities, leading to an increase in the stocking rate of the grassland instead of a decrease [11]. For instance, Dong's research demonstrates that the financial subsidy provided in GECP functions similarly to a production subsidy, indirectly contributing to an increase in the grassland stocking rate [11]. Meanwhile, Yang et al. demonstrated that following the implementation of the GECP, the proportion of grassland NDVI (greater than 0.3) in Qinghai Province decreased from 22.45% in 2011 to 19.33% in 2015 [10].

After reviewing the existing literature, a thought-provoking question arises: Why do significant variations exist in the ecological effects of the GECP across different studies? From a micro perspective, these variations may be associated with significant differences in herders' family characteristics, such as education levels, income levels, and livestock types, which can influence the ecological effects of the GECP [12,13]. In the case of families with higher levels of education, their ability to comprehend and accept policy regulations is more effective in reducing grazing intensity [5]. From a macro perspective, China possesses a vast territory with various types of grasslands. Moreover, grasslands in different regions are subject to distinct external environments [3], resulting in significant heterogeneity of grassland ecosystems in both time and space [14]. Consequently, significant regional variations may exist in the impact of the GECP on grassland ecology. However, existing research is often confined to specific areas, precluding a comprehensive understanding of the policy's overall ecological effects at a macro level, leading to controversies. However, this paper argues that the implementation effect of the policy may be affected by the grassland ecosystem vulnerability. In this paper, grassland ecosystem vulnerability is

defined as the tendency of grassland ecosystems to be adversely affected, including the sensitivity of the system itself to the natural environment and the adaptability to the internal succession of the system. So on the one hand, in areas where the grassland is relatively fragile, the ecological restoration effect of the GECP under the same incentive system may not be evident due to the weak growth ability of the grassland itself. On the other hand, in regions characterized by heightened vulnerability of grassland ecosystems, livestock production activities are vigorous, resulting in elevated grassland stocking rates. Consequently, the cost of reducing livestock per unit of grassland area is elevated compared to other regions, ultimately leading to suboptimal policy outcomes in the area.

Therefore, this paper initially assesses the vulnerability of grassland ecosystems in six pastoral provinces of China using panel data from 395 counties spanning from 2001 to 2021. Subsequently, the difference-in-difference model (DID) is employed to analyze the impact of the GECP on grassland quality. Lastly, the moderating effect model is employed to examine and discuss how the vulnerability of the grassland ecosystem moderates the impact of the GECP on the restoration of grassland vegetation. It also checks to see if the implementation effect of the GECP is different in different grassland vulnerability zones. The goal is to improve and fine-tune the grassland ecological compensation policy system that is already in place.

## 2. Materials and Methods

### 2.1. Theoretical Basis

Stimulated by the external market, the contradiction between the rapid expansion of grassland animal husbandry and the limited production capacity of grassland has been escalating, resulting in the continuous degradation of the grassland ecological environment [15]. Based on the theory of externalities and public goods, grasslands in unfenced rangeland contexts are considered typical public goods due to their rivalrous consumption and non-excludable nature [16]. Therefore, it is essential to internalize the externalities arising from the supply and consumption of the grassland ecological environment using appropriate administrative measures to improve the grassland condition [7]. The core idea of the GECP is the government's guidance to herders in reducing livestock and minimizing grassland use intensity. This aims to restore the grassland ecological environment by providing subsidies for grazing prohibition and rewards for maintaining grassland balance. Simultaneously, following the principle of beneficiary payment, the government will provide financial compensation to herders who meet the policy's requirements for reducing livestock. This compensation aims to offset the income loss resulting from the reduction in livestock [17]. In light of the above, the following hypothesis is proposed:

**H1.** *Grassland ecological compensation policy can improve grassland quality.*

Building upon the vulnerability concept introduced in the fifth report of the IPCC [18], this paper asserts that the grassland ecosystem within the highly vulnerable grassland area exhibits high sensitivity and low adaptability. Consequently, the grassland ecosystem in this area becomes more susceptible to climate impacts, among others. Meanwhile, due to ongoing changes in the external environment, the fragile grassland ecosystem in this area exhibits a deficiency in internal succession or self-renewal capability, resulting in low adaptability. For this reason, this paper argues that the ecological effect of the GECP may be regulated by the vulnerability of the grassland ecosystem. On the one hand, areas with highly fragile grasslands experience harsh climate conditions, rendering them exceptionally susceptible to adverse impacts arising from extreme climates. This will not only impact the regular growth and development of the grassland but also harm the grass tissues and organs, leading to permanent damage to its natural growth [19,20]. With the same initial stocking rate, the grassland in the highly vulnerable area exhibited weak self-growth ability, leading to no apparent grassland recovery effect.

On the other hand, regions characterized by high grassland ecosystem vulnerability experience significant human-induced disturbances. As one of the human activities most closely related to grassland ecology, animal husbandry is generally more active and developed in areas with high grassland vulnerability. Typically, regions with advanced animal husbandry exhibit industrial benefits including well-established infrastructure, advanced breeding technologies, and consistent market demand [21–23]. These factors provide favorable conditions for local herders to expand their breeding scale. In order to maximize their own interests, local herders will not hesitate to expand the scale of farming to attain higher profits. However, under the traditional grazing feeding mode, the wanton expansion of grazing scale will inevitably increase the stocking rate per unit of grassland area. According to the 2015 Grassland Monitoring Report of the Chinese government, the average livestock overloading rate in counties dominated by the pastoral economy was 18.2%, in contrast to 13.2% in other counties [24]. These data indicate a prevalently high rate of grassland livestock overloading in economically advanced pastoral regions. However, the elevation in stocking rates per unit of grassland area results in disproportionately steep costs associated with livestock reduction. Such cost escalation could erode the herders' incentive to reduce livestock numbers, possibly leading to undesirable policy outcomes in the region. Building upon the preceding theoretical analysis, the following hypothesis is proposed:

**H2.** *The positive effect of the GECP on grassland vegetation restoration will gradually decrease with the increase in grassland ecosystem vulnerability.*

### 2.2. Measurement and Classification of Grassland Ecosystem Vulnerability

Based on the previous definition of vulnerability of grassland ecosystems, it can be seen that the vulnerability of grassland ecosystems consists of two parts: sensitivity and adaptability of the system. Vulnerability is equal to the difference between sensitivity and adaptability [25]. The formula is expressed as

$$V = S - A \tag{1}$$

where $V$ is the grassland ecosystem vulnerability, S is the grassland ecosystem sensitivity, and A is the grassland ecosystem adaptability.

In this paper, grassland ecosystem sensitivity is defined as the degree of response of the grassland ecosystem to environmental change and is expressed by the interannual fluctuation in grassland ecosystem functional characteristics. Specifically, this paper uses the annual average net primary productivity (NPP) of county-level grasslands as the characteristic quantity to measure grassland ecosystem functions, and the sensitivity is represented by the interannual fluctuation in NPP from 2001 to 2021, reflecting the degree of dispersion of NPP from the average value. The calculation formula is

$$S = \frac{\sum_{i=1}^{n} \left| F_i - \overline{F} \right|}{\overline{F}} \tag{2}$$

where $i$ represents year, $F_i$ represents the annual average value of NPP, $\overline{F}$ is the average value of NPP in the study area from 2001 to 2021, and $S$ represents the variable rate of NPP and reflects the grassland ecosystem sensitivity.

Adaptability pertains to the grassland ecosystem's ability to maintain and restore its structural integrity in response to natural elements such as climate changes and other disturbances [26]. This study utilizes the slope of the linear regression trend line that illustrates the interannual variations in NPP from 2001 to 2021 to express adaptability. The calculation formula is

$$y = Ax + B \tag{3}$$

where $x$ and $A$ represent the interannual variability in NPP and the changing trend of variability; these can be calculated using the following formula [27,28]:

$$A = \frac{n \sum xy - (\sum x)(\sum y)}{n \sum x^2 - (\sum x)^2} \tag{4}$$

where $x$ represents the natural numbers 1, 2, 3, etc., corresponding to the years from 2001 to 2021, and $y$ is identified as the objective variable of NPP.

Furthermore, the natural breakpoint method in ArcGIS 10.6 was utilized to classify the susceptibility levels of grassland ecosystems in each county. Subsequently, we classified the study area into distinct categories based on the degree of grassland ecosystem vulnerability: non-vulnerable, slightly vulnerable, moderately vulnerable, severely vulnerable, and extremely vulnerable areas.

### 2.3. Econometric Model

(1) Baseline model. For this paper, the difference-in-difference (DID) method was used to estimate GECP's grassland quality. This was accomplished based on the program implementation areas outlined in the 2011 Guidance on the Implementation of the Grassland Ecological Protection Subsid. Specifically, whether the program was implemented in 2011 is the treatment variable in this paper; the treatment group is the 338 counties in the six Chinese pastoralist provinces that implemented the program, and the control group is the 57 counties that have not implemented the policy in the six pastoral provinces. For the time variable, after 2011, the value is 1; otherwise, it is 0. The estimation model is as follows:

$$Y_{it} = \beta_0 + \beta_1 treat \times time + \theta X_{it} + a_i + \mu_t + \varepsilon_{it} \tag{5}$$

where $i$ and $t$ represent the county and year, and the dependent variable $Y_{it}$ represents the grassland quality, indicated by the average net primary productivity (NPP) of grassland across individual counties. The coefficient $\beta_1$ of the interaction term between treatment variables and time variables is the concern of this paper and is expressed as the average treatment effect of the GECP on grassland quality. $a_i$ is the individual fixed effect, $\varepsilon_{it}$ is the time fixed effect, and $\varepsilon_{it}$ is the random error term. $X_{it}$ is a set of control variables, including both natural and socioeconomic factors.

(2) Dynamic effect model. In order to test the hypothesis of the parallel trend of DID and to further examine the dynamic changes in the grassland quality in response to the GECP, for this paper, the following model is constructed with reference to the Event Study Approach proposed by Jacobson et al. [29]:

$$Y_{it} = \beta_0 + \Sigma_{t=2001}^{2021} \delta_t treat \times time_{it} + \theta X_{it} + a_i + \mu_t + \varepsilon_{it} \tag{6}$$

where $\delta_t$ denotes a series of estimates from 2001 to 2021, and the other variables are the same as in the baseline regression.

(3) Moderating effect model. In order to further identify the moderating effect of grassland ecosystem vulnerability on GECP affecting grassland quality, the following test model is constructed:

$$Y_{it} = \beta_0 + \beta_1 treat \times time + \beta_2 treat \times time \times V_{it} + \theta X_{it} + a_i + \mu_t + \varepsilon_{it} \tag{7}$$

where $V$ represents the vulnerability of the grassland ecosystem; the coefficient $\beta_2$ can reflect the adjustment effect produced by the vulnerability of the grassland ecosystem, and the other variables are the same as in the baseline regression.

### 2.4. Variable Selection

Dependent variable. In essence, the core of the GECP is to directly or indirectly restore and improve grassland productivity by controlling the number of livestock animals, so as to realize the sustainable development of the grassland ecosystem. In this paper, grassland

quality specifically refers to grassland productivity, which is defined as the amount of material products produced by a unit area of grassland within a certain period of time under certain input conditions. Since the net primary productivity (NPP) is the total amount of organic matter accumulated by green plants per unit time and unit area, which can directly reflect the growth of vegetation [30,31], this paper uses NPP as a measure of grassland quality.

Control variables. Referring to previous research results [2,32], the control variables are selected from two aspects: nature and the social economy. Specifically, we include annual average precipitation (P), annual average temperature (T), and snow cover days (SCD) at the county level to prevent the influence of natural climatic factors. These data come from the China National Meteorological Information Center (http://data.cma.cn/, accessed on 8 August 2023) and the National Glacial Permafrost Desert Scientific Data Center (http://www.ncdc.ac.cn/, accessed on 8 August 2023). Concurrently, factors such as animal husbandry development level (AHDL), industrial structure (IS), urbanization level (URB), and government scale (GS) are selected to reflect the influence of social and economic factors. Specifically, the ratio of the added value of the primary industry to GDP is used to measure the level of animal husbandry development, the ratio of the added value of the secondary industry to GDP is used to measure the industrial structure, the ratio of the non-agricultural population to the total population is used to measure the level of urbanization, and the ratio of county-level fiscal expenditure to GDP is used to measure the size of the government. The above data are all sourced from the "China County (Province) Socio-Economic Statistical Yearbook" and corresponding statistical bulletins. The descriptive statistics of the main variables are shown in Table 1.

**Table 1.** Descriptive statistics of the main variables.

| Variable | Observations | Mean | Min | Max | Std. Dev. |
|---|---|---|---|---|---|
| NPP (gC m$^{-2}$ a$^{-1}$) | 8295 | 286.252 | 11.162 | 871.441 | 195.119 |
| P (mm × 10$^3$) | 8295 | 0.491 | 0.007 | 2.778 | 0.335 |
| T (°C) | 8295 | 7.204 | −8.416 | 20.526 | 6.585 |
| SCD (day) | 8295 | 39.152 | 0 | 213 | 44.263 |
| AHDL (%) | 8295 | 0.151 | 0.081 | 0.309 | 0.042 |
| IS (%) | 8295 | 0.429 | 0.204 | 0.584 | 0.072 |
| URB (%) | 8295 | 0.403 | 0.224 | 0.633 | 0.126 |
| GS (%) | 8295 | 0.368 | 0.113 | 1.379 | 0.304 |

## 3. Results and Discussion

### 3.1. Vulnerability Distribution Patterns of Grassland Ecosystems

Utilizing the natural breakpoint method, we categorize the evaluation results of grassland ecological vulnerability into five levels: non-vulnerable (−7.599~−4.130), slightly vulnerable (−4.130~−1.398), moderately vulnerable (−1.398~0.021), severely vulnerable (0.021~1.257), and extremely fragile (1.257~3.305). Overall, the vulnerability of grassland ecosystems in the study area showed an obvious spatial differentiation pattern. The northeastern and northwestern regions displayed the highest vulnerability, followed by the central region, while the southwestern area exhibited the lowest vulnerability (Figure 1). Specifically, the regions with severely and extremely vulnerable grasslands are predominantly clustered in Inner Mongolia, Ningxia, Gansu, and the northwestern Xinjiang cities of Karamay and Urumqi. The moderately vulnerable areas are primarily situated across Qinghai, Sichuan, and eastern Tibet, and the lightly vulnerable and non-vulnerable areas are mainly concentrated in southern Xinjiang and western Tibet.

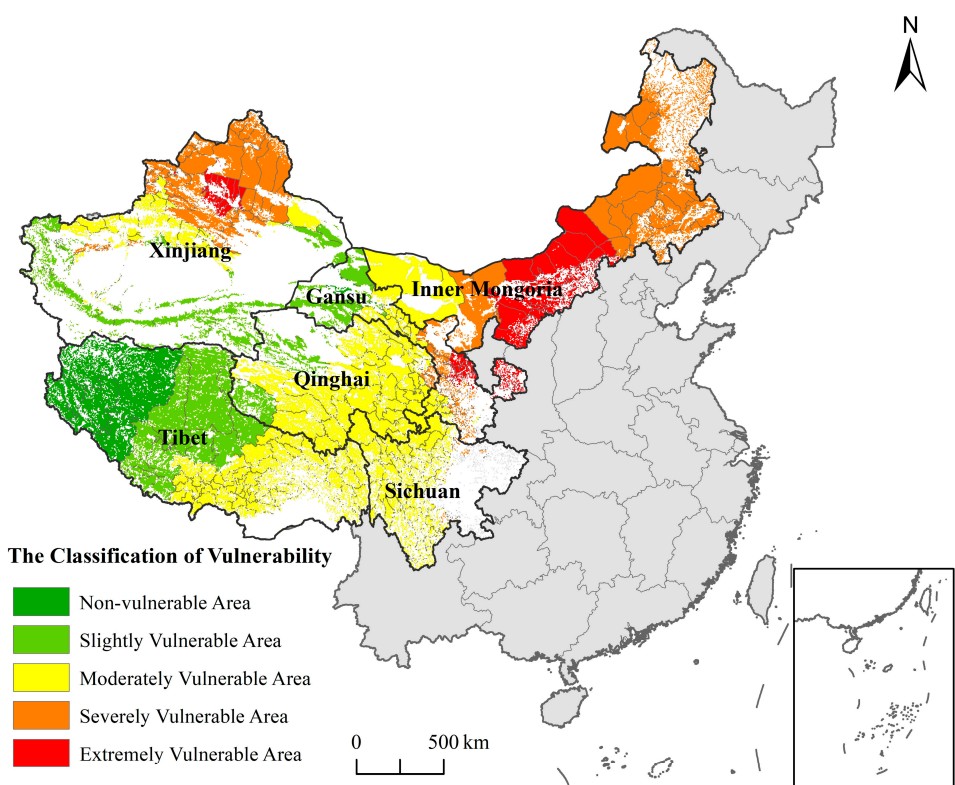

**Figure 1.** The classification of grassland ecosystem vulnerability.

After categorizing the research area based on the vulnerability of the grassland ecosystem, it can be found that the areas with higher grassland vulnerability tend to have harsh natural environments and are easily disturbed by human activities, which is consistent with the previous theoretical conjecture. In particular, alpine desert grasslands, temperate desert grasslands, and temperate steppe deserts are the most common types of grasslands in areas that are extremely or severely vulnerable. This is based on the natural conditions of the grasslands and how they are divided in the Chinese grassland classification system [33,34]. The water and heat conditions in these areas are poor, and grassland growth cannot obtain enough water, resulting in a decline in grassland productivity. In addition, wind erosion and water erosion are also the main factors restricting the growth of grassland in this area. Wind erosion and water erosion will wash away soil and vegetation, making the land barren and unsuitable for plant growth. However, in other vulnerable grassland regions, tussock and meadow dominate the grassland types, offering comparatively favorable growth conditions. According to the grade evaluation of grassland resources in China, although the distribution of grasslands in China is not strong in the longitudinal and latitudinal directions, the quality of grasslands in general shows a pattern of gradual decline from east to west and from south to north [35,36], which is also consistent with that in this paper. The assessment results of grassland ecosystem vulnerability confirmed each other.

In addition, in terms of the impact of human activities, in areas with high grassland vulnerability, human disturbance activities such as animal husbandry production are also relatively intensive and active. It can be seen from Figure 1 that the extremely vulnerable areas, moderately vulnerable areas, and non-vulnerable areas of grassland are concentrated in Inner Mongolia, Qinghai, and Tibet, respectively, and their output values of animal husbandry were CNY 175.53 billion, CNY 29.86 billion, and CNY 12.93 billion, respectively. This suggests a potentially substantial correlation between the evolution of the animal husbandry economy and the vulnerability of grassland ecosystems. In China, a notable trend emerges: in areas where the scale of livestock development is greater, the local grassland ecosystem may be more vulnerable.

### 3.2. Impact on Grassland Quality: Baseline Regression Results

Table 2 reports the baseline regression results for the effects of the GECP on grassland quality. In this table, column (1) has no control variables, and columns (2) and (3) are the regression results after incorporating natural factors and economic and social factors in turn, while controlling the time and regional fixed effects. Regardless of the model, the GECP has a positive impact on grassland quality. Controlling for all variables, the coefficient between the GECP and grassland quality is 0.033, and it passed the significance test at the 1% level. This implies that implementing the grassland ecological compensation policy can lead to an average 3.3% increase in the net primary productivity (NPP) of grassland within the policy area; H1 is thus verified. This finding closely aligns with Hou et al.'s [2] discovery that the implementation of the grassland ecological compensation policy resulted in a 3.2% rise in NDVI. This congruence adds a supplementary layer of validation to the outcomes of our research.

**Table 2.** Effects of the GECP on grassland quality: 2001–2021.

| Variable | ln(NPP) | | |
|---|---|---|---|
| | **(1)** | **(2)** | **(3)** |
| treat × time | 0.100 *** | 0.082 *** | 0.031 *** |
| | (42.516) | (34.844) | (9.113) |
| P (mm × $10^3$) | | 0.112 *** | 0.063 *** |
| | | (26.237) | (15.208) |
| T (°C) | | 0.019 *** | 0.008 *** |
| | | (10.767) | (4.637) |
| SCD (%) | | −0.011 *** | −0.003 * |
| | | (−6.065) | (−1.736) |
| AHDL (%) | | | 0.061 *** |
| | | | (11.603) |
| IS (%) | | | −0.038 |
| | | | (−0.573) |
| URB (%) | | | −0.127 *** |
| | | | (−4.597) |
| GS (%) | | | 0.004 *** |
| | | | (15.220) |
| Constant | 5.329 *** | 4.648 *** | −0.191 *** |
| | (3707.013) | (165.033) | (126.314) |
| County fixed effects | No | Yes | Yes |
| Year fixed effects | No | Yes | Yes |
| Observations | 8295 | 8295 | 8295 |
| R-squared | 0.202 | 0.467 | 0.512 |

Notes: Values outside of brackets are the parametric estimation values; values in brackets are *t*-test values; * and *** represent significance at 10% and 1%, respectively.

However, our study is different because we used the county's mean grassland NPP as a measure of grassland quality instead of using the average value of all vegetation in the county administrative zone as a stand-in variable. The central emphasis of the GECP indisputably resides within the realm of the grassland biome, demarcated from other land categorizations. Therefore, utilizing the average net primary productivity (NPP) of grassland across the county presents a more informative avenue to unveil the genuine repercussions of this policy. Secondly, given the potential for substantial heterogeneity between provinces with and without policy implementation, this study employs counties within policy-implemented provinces that have refrained from policy implementation as the control group. This approach diverges from using counties in unrelated provinces lacking policy implementation. The above approach can also make up for previous research deficiencies. In order to further test the authenticity of the results in this paper, a series of robustness tests were carried out as follows.

### 3.3. Robustness Tests

3.3.1. Parallel Trend Test and Dynamic Effect Analysis

The differences in dependent variables between different groups may not only come from the treatment effect of the policy but may also involve the influence of other confounding factors. Based on this, this paper considers the estimation results of $\delta_t$ of grassland NPP in the four years before and after the implementation of the GECP under the 95% confidence interval (as shown in Figure 2). Clearly, before the policy's implementation, the coefficient $\delta_t$ lacks statistical significance. This implies that the selected variables effectively untangle the policy's impact from external influences and better satisfy the parallel trend assumption. After the policy was implemented, the coefficient $\delta_t$ exhibited a notably positive significance, signifying the efficacious role of the GECP in fostering the recuperation of grassland vegetation.

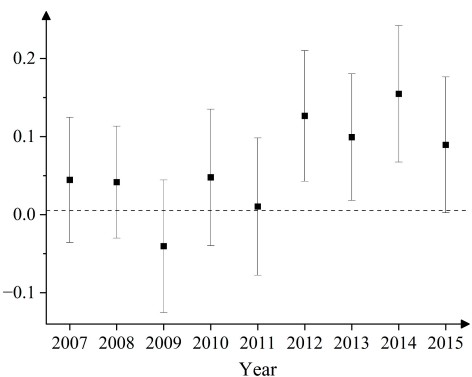

**Figure 2.** Parallel trend and dynamic effect test.

3.3.2. Replacing the Explanatory Variables

The county-level grassland vegetation coverage index (NDVI) was selected as the explained variable for regression verification. According to column (1) of Table 3, the GECP has a stable effect on grassland quality, and it can be generally proved that the compensation policy is conducive to the restoration of grassland vegetation.

**Table 3.** Robustness test.

| Variable | (1) | (2) | (3) |
| --- | --- | --- | --- |
| | **NDVI** | **ln(NPP)** | **ln(NPP)** |
| treat × time | 0.016 *** | 0.083 *** | 0.031 *** |
| | (12.303) | (6.152) | (2.606) |
| Constant | 0.372 *** | 4.495 *** | 4.280 *** |
| | (23.216) | (17.627) | (31.328) |
| Control variables | Yes | Yes | Yes |
| County fixed effects | Yes | Yes | Yes |
| Year fixed effects | Yes | Yes | Yes |
| Observations | 8295 | 8295 | 868 |
| R-squared | 0.519 | 0.698 | 0.396 |

Notes: Values outside of brackets are the parametric estimation values; values in brackets are *t*-test values; *** represents significance at 1%.

3.3.3. Adding City and Year Fixed Effects

In order to control for grassland quality possibly being affected by city-level characteristic factors and event impacts, a dummy variable was constructed for the interaction between city and year and added to the baseline regression. The results in column 2 of Table 3 showed that after incorporating interactive variables, the regression coefficient was still significantly positive, indicating that the GECP was beneficial to improving grassland quality.

### 3.3.4. Replace Data Range

Furthermore, for this paper, the existing county-level data were replaced with the corresponding city-level data in an attempt to test the robustness of the benchmark regression from a larger scale. As indicated in column (3) of Table 3, there is no significant difference with the benchmark regression results, which shows the robustness of the benchmark regression results in this paper.

### 3.3.5. Placebo Test

In addition to the GECP, other policy factors and unobservable factors may have an impact on the explained variables, potentially leading to skewed estimation outcomes. This study employs placebo tests and the difference-in-differences (DID) model to establish a simulated environment and quasi-natural experiment, aiming to assess potential risks. Specifically, an experimental group is randomly generated for the annual GECP by the random sampling method, and then the estimated value of the coefficient of the difference-in-difference term is wrong. As can be seen from Figure 3, the t values of most random sampling results are near zero, which once again verifies the robustness of the benchmark regression results.

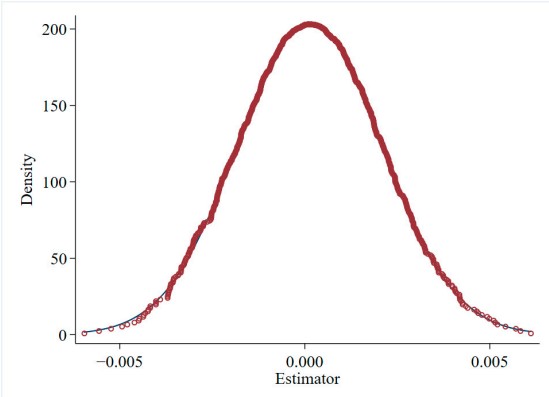

**Figure 3.** Placebo test.

### 3.4. Moderating Effect Analysis

The empirical results of the moderating effect on grassland ecosystem vulnerability are shown in Table 4. The coefficient $\beta_2$ for the interaction term is significantly negative at the 1% level. Meanwhile, the coefficient $\beta_1$ for the policy variable shows significant positivity. This implies that the vulnerability of the grassland ecosystem has partially restrained the beneficial impact of the GECP on grassland quality; thus, H2 is verified. Specifically, this suggests that the policy's positive effects diminish as the grassland ecosystem's vulnerability rises. When this result is combined with the spatial distribution of grassland ecosystem vulnerability, the degree of the positive impact of the GECP on grassland quality in different regions can be drawn, as shown in Figure 4. Furthermore, according to the classification of grassland vulnerability, group regression was performed on the benchmark regression, and the results are shown in Table 5. It can be seen that there are significant differences in the significance of the coefficient $\beta_1$ of policy variables on grassland quality in different groups. Specifically, the positive impact of policies on grassland quality changed from significant to insignificant with the increase in grassland ecosystem vulnerability. This effect is not obvious in severely and extremely vulnerable areas.

**Table 4.** The moderating effect of grassland ecosystem vulnerability.

| Variable | ln(NPP) | |
|---|---|---|
| | **Coefficient Estimates** | **t-Value** |
| treat × time | 0.164 *** | 24.542 |
| treat × time × V | −0.055 *** | −22.939 |
| P (mm × $10^3$) | 0.083 *** | 19.676 |
| T (°C) | 0.011 *** | 6.759 |
| SCD (day) | −0.003 * | −1.797 |
| AHDL (%) | −0.542 *** | −9.264 |
| IS (%) | −0.185 *** | −6.850 |
| URB (%) | 0.003 *** | 11.227 |
| GS (%) | −0.117 *** | −10.244 |
| Constant | 4.891 *** | 126.495 |
| County fixed effects | Yes | |
| Year fixed effects | Yes | |
| Observations | 8295 | |
| R-squared | 0.469 | |

Notes: Values outside of brackets are the parametric estimation values; values in brackets are *t*-test values; * and *** represent significance at 10% and 1%, respectively.

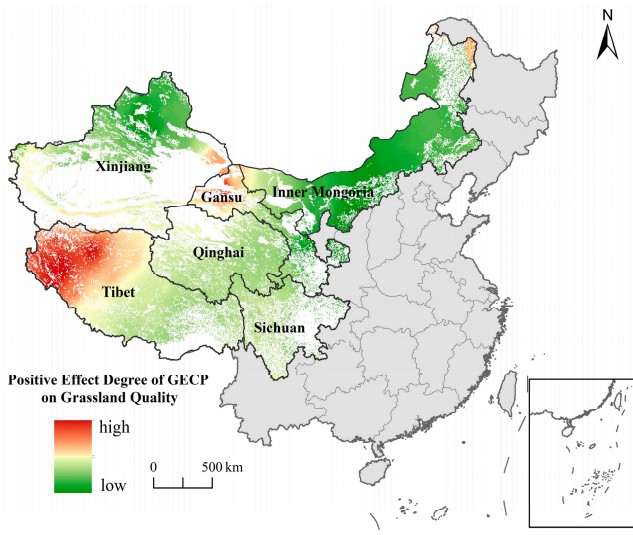

**Figure 4.** Distribution of the degree of influence of the GECP on grassland quality in different regions.

**Table 5.** Effects of the GECP on grassland quality under different grassland vulnerability zones.

| Variable | ln(NPP) | | | | |
|---|---|---|---|---|---|
| | **(1)** | **(2)** | **(3)** | **(4)** | **(5)** |
| | **Extremely** | **Severely** | **Moderately** | **Slightly** | **Non** |
| treat × time | 0.005 | 0.001 | 0.029 *** | 0.047 *** | 0.075 *** |
| | (0.820) | (0.202) | (9.163) | (6.076) | (4.869) |
| Constant | 4.692 *** | 4.143 *** | 5.427 | 3.121 *** | 3.492 *** |
| | (23.700) | (41.089) | (144.392) | (49.351) | (35.774) |
| Control variables | Yes | Yes | Yes | Yes | Yes |
| County fixed effects | Yes | Yes | Yes | Yes | Yes |
| Year fixed effects | Yes | Yes | Yes | Yes | Yes |
| Observations | 1554 | 2436 | 3360 | 777 | 168 |
| R-squared | 0.878 | 0.626 | 0.267 | 0.514 | 0.484 |

Notes: Values outside of brackets are the parametric estimation values; values in brackets are *t*-test values; *** represents significance at 1%.

Based on the above regression results, this paper suggests that the vulnerability of the grassland ecosystem can adjust the impact of the GECP on grassland quality. On the one hand, from the perspective of natural factors, grasslands are located in areas with extremely or severely vulnerable grasslands, grassland desertification is serious, and the grassland experiences unfavorable site conditions and possesses limited inherent growth capability. Under the same policy incentives, regions characterized by elevated grassland ecosystem vulnerability exhibit limited natural regrowth capacities. This leads to the inconspicuous restoration of grassland vegetation and ultimately makes the policy effect less effective than that in other areas. On the other hand, from the perspective of human disturbance, as can be seen from the above, the vulnerability of the grassland ecosystem is inversely correlated with the intensity of livestock production activities. This environment provides favorable conditions for the formation of the whole industry chain of animal husbandry in this region. And it not only improves the level of breeding technology in this region, but also promotes the stability of animal husbandry market demand. Therefore, under the same conditions, herders in this area are more likely to expand their farming scale compared to others. However, this is also a dangerous signal. First, herders aim to expand breeding operations to generate higher returns, driven by ongoing market incentives. Local herders may take advantage of the industrial advantages to facilitate easier expansion of breeding scale, resulting in a higher stocking rate of the local grassland compared to other areas. But at the same time, the cost of livestock reduction per unit of grassland area will gradually increase with the increase in stocking rate. Under the same policy incentive, the effect of the policy in this area will not be obvious due to the high cost of livestock reduction. Second, related research shows that in the current context characterized by generally low compensation standards, some herders not only refrain from reducing livestock as required but instead utilize compensation funds to expand their breeding operations [37,38]. Similarly, herders in this region can perform such "violation" practices more efficiently and conveniently, resulting in an adverse impact on the policy's positive effect.

In contrast to previous studies, Lin et al.'s study demonstrates substantial variation in how diverse grassland types respond to the GECP [14]. By contrasting the net primary productivity (NPP) of each grassland type prior to and after policy implementation, the study reveals that the NPP alteration can reach 135.19% in the temperate forest grassland, contrasting with less than 50% in other regions. This indicates that the GECP exhibits a more pronounced efficacy within the temperate forest grassland, predominantly situated in western Tibet. Notably, this region closely corresponds to the non-vulnerable grassland ecosystem discussed in this study, thereby offering further endorsement to the empirical findings presented in this paper. In addition, some micro-empirical studies have shown that some livestock reduction subsidies and grazing prohibition subsidies have no effective effect on reducing the number of local livestock animals, especially in the eastern and central regions of Inner Mongolia [1,8,39]. Moreover, this region coincides precisely with the concentration of highly vulnerable or severely vulnerable grassland areas, where the impact of policy implementation is not conspicuously evident. This alignment further strengthens the credibility of the research findings.

## 4. Conclusions

The grassland ecological compensation policy serves as an important tool for alleviating grassland carrying capacity and reversing grassland degradation. However, considering that the implementation effects of the GECP in different regions are quite different in reality, this paper innovatively incorporates the vulnerability of grassland ecosystems into the analysis framework of the policy. And it discusses the role of grassland ecosystem vulnerability between GECP and grassland quality at the regional level, in order to more accurately and objectively evaluate the ecological performance of the policy. Moreover, this not only helps to adjust and optimize existing policies at the regional level, but also has important theoretical and practical significance for improving the grassland ecological compensation mechanism and maintaining the balance of the grassland ecosystem.

This paper uses a county-level panel dataset that includes 395 counties in six provinces in China's main pastoral regions from 2001 to 2021 to test the effect of grassland ecological compensation policy on grassland quality and the effect that grassland ecosystem vulnerability has on that effect. The main findings of this paper are as follows: (1) The spatial distribution of grassland ecosystem vulnerability across the six provinces within China's pastoral regions exhibits discernible differentiation. The severely and extremely vulnerable areas of grassland are mainly concentrated in northern regions such as Inner Mongolia and Xinjiang, while the slightly vulnerable and non-vulnerable areas are distributed in southwestern regions such as Tibet. (2) The GECP has a significant positive impact on grassland quality as a whole, which shows that the GECP is conducive to the improvement of grassland quality. (3) Grassland ecosystem vulnerability exerts a negative regulatory influence on the relationship between the GECP and grassland quality. As the degree of grassland ecosystem vulnerability increases, the positive effect of the GECP on grassland quality progressively diminishes.

The above findings have the following implications for improving the grassland ecological compensation policy and optimizing the ecological compensation system: (1) Moderately increase compensation standards. The root cause of the policy's failure to achieve desired results is that compensation funds are not effective enough to make up for loss costs. More herders will choose illegal methods such as stealing grazing and night grazing and expand the scale of grazing to maximize their benefits. Therefore, it is necessary to moderately increase the compensation standard to reduce losses of herders, thereby encouraging herders to reduce livestock to restore grassland, and necessary supervision may make this effect more significant. (2) Implement differentiated subsidies. Substantial variations exist in the natural and economic contexts across diverse regions. Grassland growth tends to be weak in areas of high grassland ecosystem vulnerability. At the same time, animal husbandry is often developed in these areas, and the stocking rate per unit area is high, resulting in a significantly higher opportunity cost of reducing livestock than that in other areas. Therefore, policies should increase subsidies for areas with relatively high vulnerability of grassland ecosystems. Of course, there are some shortcomings in this study. This paper starts from the perspective of the vulnerability of grassland ecosystems, it responds to the controversy over the ecological effects of the GECP. However, because of the policy's extensive reach, research from existing perspectives may not fully explain the heterogeneous effects of the policy. Given the significant variations in economic development and national culture across different regions, subsequent studies must concentrate on analyzing these dimensions. Moreover, delving into the root causes of this heterogeneity is also essential to maximize the intended impact of the policy.

**Author Contributions:** Conceptualization, M.L. and W.W.; methodology, M.L. and W.W.; software, M.L. and W.W.; validation, M.L. and H.L.; formal analysis, M.L.; investigation, M.L.; resources, M.L.; data curation, M.L.; writing—original draft preparation, M.L. and W.W.; writing—review and editing, M.L. and W.W.; visualization, M.L.; supervision, H.L.; project administration, H.L.; funding acquisition, H.L. All authors have read and agreed to the published version of the manuscript.

**Funding:** This research was funded by the Shaanxi Provincial Natural Science Project, "Research on the Foreign Exchange Increase Effect of Shaanxi State-owned Forest Farm Investment and Its Enhancement Strategy" (S2023-JC-YB-2373).

**Institutional Review Board Statement:** Not applicable.

**Informed Consent Statement:** Not applicable.

**Data Availability Statement:** The data presented in this study are available in this article.

**Conflicts of Interest:** The authors declare no conflict of interest.

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
