# Peer review of "The Influence of Grassland Ecological Compensation Policy on Grassland Quality: Evidence from the Perspective of Grassland Ecosystem Vulnerability"

_agriculture, doi:10.3390/agriculture13091841_

Round 1

Reviewer 1 Report

Thanks for the opportunity to review this paper, which examines the effectiveness of a grassland grazing compensation scheme across different areas of China.

Overall the paper is good, the topic is important, and I think it should be published after some minor alterations to improve its clarity and impact.

I have three main issues which need to be addressed:

1)      The figures should be re-organised to focus on the paper’s main results. Currently, the visual figures are focussed on the robustness tests, while the main findings of the paper are mostly expressed in tables and statistics. I would prefer to see Figure 3 removed (Placebo test) and replaced with a figure that expresses the main findings.

2)      There needs to be more discussion about how grassland “condition” (and vulnerability) is conceived and defined. Currently this is not clear enough. First, it must be clear that grassland quality (in this paper) relates to grass cover / greenness and productivity. It does not include other possible measures such as plant species richness, or the presence or abundance of rare or grazing-sensitive species (measures which are sometimes used to assess grassland condition in other contexts, and which are important to some readers such as plant ecologists). Second, it must be clearer how the concept of condition relates to the natural capacity of the area (for example- does a dessert with naturally low cover have naturally low condition, or does the metric adjust for this? The statement “These regions exhibit significant grassland desertification, accompanied by unfavorable site conditions and limited growth potential” is an example of where this is unclear. Is desertification and limited growth potential a problem in a natural desert?). This is currently expressed in the formulae, but it is not explained clearly enough in the text. Third, it must be clear what the original metrics of the GCEP scheme actually are: Does the scheme aim to impact NPP? NDVI? What is its measure? This is currently not clear, but is important for fair evaluation.

3)      It should be more clear which statements are direct interpretation of the study data, and which statements are un-tested hypotheses offered as discussion. Currently, there are two examples of statements which are not directly shown in the data (1) line 295: “Presently in China, a notable trend emerges: as the scale of the animal husbandry sector expands, the vulnerability of the grassland ecosystem appears to increase.”  2) Line 411: ““in the current context characterized by generally low compensation standards, some herders not only refrain from reducing live-stock as required but instead utilize compensation funds to expand their breeding operations”.). These are OK- I don’t disagree with them- but I think it should be clearer that they are suppositions not drawn from the data.

I also noted the following minor problems which should be addressed:

Line 12: Remove double negative (“reverse decline of degradation”)

Line 13: “healthy and stable development of the grassland ecosystem”. Do you mean economic development?

Line 34: By 2020? I think this date is wrong.

Line 66: make it clear what “panel data” is. Not all readers will know.

Line 70: define NDVI.

Line 116: In considering whether grasslands are public goods, it is best to be more specific and say that “grasslands held in common”, or “grasslands in unfenced rangeland contexts” are public goods. The term “grassland” refers to an ecosystem type, and it may not always be a public good as it is in China (e.g. In Europe and Australia most grassland is fenced and used for private livestock ventures).

Line 120: “sustain its ecological balance:” What does this mean? How does this relate to vulnerability or condition?

Line 128: “vegetation restoration”: what does this mean, and how does it relate to vulnerability or condition?

Line 146: meaning of words “should be” is unclear: is this a prediction of what does occur, or a recommendation for what you want to occur?

Line 178: Define NPP.

Line 208: Not quite clear from the grammar how the missing data is treated here. Write more clearly.

Line 253: An awkward start to the sentence: “Specifically, firstly,…”

Section 4.3.4 (from line 352) is too vague: More detail is required for me to understand what has been done here, and what it means.

Line 367: Is this Figure 3 or Figure 4??

Line 377: Re-write confusing phrase “significantly negatively significant”

Line 389” Change “This paper believes” to “this paper suggests” or “we believe” (the paper itself does not have beliefs).

Line 466: Be clear what you mean by “the policy failure”.

Line 469: Do you really believe that higher compensation will decrease dishonesty? I doubt this! Surely more inspection / check-ups would help more.

The quality of English is moderate, and would benefit from proof reading from an English speaker. Generally, the meaning is clear, but the expression is sometimes a little bit awkward. I have noted in my comments some of the few places where the expression is not clear.

I appreciate this is difficult for those who do not speak English as a first language.

(I am an English speaker)

Author Response

Point 1: The figures should be re-organised to focus on the paper’s main results. Currently, the visual figures are focussed on the robustness tests, while the main findings of the paper are mostly expressed in tables and statistics. I would prefer to see Figure 3 removed (Placebo test) and replaced with a figure that expresses the main findings.

Response 1: Thanks for the reviewer’s kind suggestion. The main finding of this paper is that there is regional heterogeneity in the impact of GECP on grassland quality, so Figure 4 was drawn to intuitively reflect it. The revised details can be found in the red font section on lines 506-507. At the same time, in most papers, the placebo test results are expressed in the form of Figure 3, so Figure 3 is retained in this paper. If the reviewers have better suggestions, this article will be further revised.

Point 2: There needs to be more discussion about how grassland “condition” (and vulnerability) is conceived and defined. Currently this is not clear enough. First, it must be clear that grassland quality (in this paper) relates to grass cover / greenness and productivity. It does not include other possible measures such as plant species richness, or the presence or abundance of rare or grazing-sensitive species (measures which are sometimes used to assess grassland condition in other contexts, and which are important to some readers such as plant ecologists). Second, it must be clearer how the concept of condition relates to the natural capacity of the area (for example- does a dessert with naturally low cover have naturally low condition, or does the metric adjust for this? The statement “These regions exhibit significant grassland desertification, accompanied by unfavorable site conditions and limited growth potential” is an example of where this is unclear. Is desertification and limited growth potential a problem in a natural desert?). This is currently expressed in the formulae, but it is not explained clearly enough in the text. Third, it must be clear what the original metrics of the GCEP scheme actually are: Does the scheme aim to impact NPP? NDVI? What is its measure? This is currently not clear, but is important for fair evaluation.

Response 2: Thanks for the reviewer’s kind suggestion. We have made correction according to the reviewer's Points. First, this paper clarifies the concept of grassland ecosystem vulnerability. The revised details can be found in the red font section on lines 105-108. Secondly, in the dependent variable selection section, the concept of grassland quality is clearly given. The revised details can be found in the red font section on lines 261-268. Finally, this article also provides a detailed explanation of the relationship between grassland ecosystem vulnerability and local ecological condition. The revised details can be found in the red font section on lines 317-330.

Point 3: It should be more clear which statements are direct interpretation of the study data, and which statements are un-tested hypotheses offered as discussion. Currently, there are two examples of statements which are not directly shown in the data (1) line 295: “Presently in China, a notable trend emerges: as the scale of the animal husbandry sector expands, the vulnerability of the grassland ecosystem appears to increase.” 2) Line 411: ““in the current context characterized by generally low compensation standards, some herders not only refrain from reducing live-stock as required but instead utilize compensation funds to expand their breeding operations”.). These are OK- I don’t disagree with them- but I think it should be clearer that they are suppositions not drawn from the data.

Response 3: Thanks for the reviewer’s kind suggestion. We have made correction according to the reviewer’s Points. The revised details can be found in the red font section on lines 345-347, lines 484-487.

Point 4: Line 12: Remove double negative (“reverse decline of degradation”)

Response 4: Thanks for the reviewer’s kind suggestion. We have made correction according to the reviewer’s Points. The revised details can be found in the red font section on lines 12-13.

Point 5: Line 13: “healthy and stable development of the grassland ecosystem”. Do you mean economic development?

Response 5: Thanks for the reviewer’s kind suggestion. We have made correction according to the reviewer’s Points. The revised details can be found in the red font section on lines 12-13.

Point 6: Line 34: By 2020? I think this date is wrong.

Response 6: Thanks for the reviewer’s kind suggestion. We have made correction according to the reviewer’s Points. The revised details can be found in the red font section on line 39.

Point 7: Line 66: make e it clear what “panel data” is. Not all readers will know.

Response 7: Thanks for the reviewer’s kind suggestion. We have made correction according to the reviewer’s Points. The revised details can be found in the red font section on lines 75-78.

Point 8: Line 70: define NDVI.

Response 8: Thanks for the reviewer’s kind suggestion. We have made correction according to the reviewer’s Points. The revised details can be found in the red font section on line 80.

Point 9: Line 116: In considering whether grasslands are public goods, it is best to be more specific and say that “grasslands held in common”, or “grasslands in unfenced rangeland contexts” are public goods. The term “grassland” refers to an ecosystem type, and it may not always be a public good as it is in China (e.g. In Europe and Australia most grassland is fenced and used for private livestock ventures).

Response 9: Thanks for the reviewer’s kind suggestion. We have made correction according to the reviewer’s Points. The revised details can be found in the red font section on lines 132-133.

Point 10: Line 120: “sustain its ecological balance:” What does this mean? How does this relate to vulnerability or condition?

Response 10: Thanks for the reviewer’s kind suggestion. We have made correction according to the reviewer’s Points. The revised details can be found in the red font section on line 136.

Point 11: Line 128: “vegetation restoration”: what does this mean, and how does it relate to vulnerability or condition?

Response 11: Thanks for the reviewer’s kind suggestion. We have made correction according to the reviewer’s Points. The revised details can be found in the red font section on lines 145-146.

Point 12: Line 146: meaning of words “should be” is unclear: is this a prediction of what does occur, or a recommendation for what you want to occur?

Response 12: Thanks for the reviewer’s kind suggestion. We have made correction according to the reviewer’s Points. The revised details can be found in the red font section on line 164.

Point 13: Line 178: Define NPP.

Response 13: Thanks for the reviewer’s kind suggestion. We have made correction according to the reviewer’s Points. The revised details can be found in the red font section on lines 199-200.

Point 14: Line 208: Not quite clear from the grammar how the missing data is treated here. Write more clearly.

Response 14: Thanks for the reviewer’s kind suggestion. We have made correction according to the reviewer’s Points. The revised details can be found in the red font section on lines 232-236.

Point 15: Line 253: An awkward start to the sentence: “Specifically, firstly,…”

Response 15: Thanks for the reviewer’s kind suggestion. We have made correction according to the reviewer’s Points. The revised details can be found in the red font section on lines 285-297.

Point 16: Section 4.3.4 (from line 352) is too vague: More detail is required for me to understand what has been done here, and what it means.

Response 16: Thanks for the reviewer’s kind suggestion. We have made correction according to the reviewer’s Points. The revised details can be found in the red font section on lines 414-416.

Point 17: Line 367: Is this Figure 3 or Figure 4??

Response 17: Thanks for the reviewer’s kind suggestion. We have made correction according to the reviewer’s Points. The revised details can be found in the red font section on line 427.

Point 18: Line 377: Re-write confusing phrase “significantly negatively significant”

Response 18: Thanks for the reviewer’s kind suggestion. We have made correction according to the reviewer’s Points. The revised details can be found in the red font section on lines 443-444.

Point 19: Line 389” Change “This paper believes” to “this paper suggests” or “we believe” (the paper itself does not have beliefs).

Response 19: Thanks for the reviewer’s kind suggestion. We have made correction according to the reviewer’s Points. The revised details can be found in the red font section on line 460.

Point 20: Line 466: Be clear what you mean by “the policy failure”.

Response 20: Thanks for the reviewer’s kind suggestion. We have made correction according to the reviewer’s Points. The revised details can be found in the red font section on lines 544-545.

Point 21: Line 469: Do you really believe that higher compensation will decrease dishonesty? I doubt this! Surely more inspection / check-ups would help more.

Response 21: Thanks for the reviewer’s kind suggestion. We have made correction according to the reviewer’s Points. The revised details can be found in the red font section on lines 547-550.

Reviewer 2 Report

This is an interesting research analyzing the impact of grassland compensation policies on enhancing grassland quality by employing a county-level panel dataset from 2001 to 2021. A theoretical background provides the basis for defining the main hypothesis of the study. The vulnerability of the grassland ecosystem has been computed, and the association between GECP and grassland quality have been empirically tested using the difference-in-difference (DID), Dynamic effect model, and moderating effect models. The study has shown that the grassland ecological compensation policy had a positive impact on grassland quality and grassland ecosystem vulnerability.  The spatial differentiation pattern of grassland ecosystem vulnerability is well explained.  The positive impacts of policies become more pronounced as the vulnerability of grassland ecosystems decreases. The main conclusions from the study are well-articulated with emphasis on the spatial distribution of grassland ecosystem vulnerability, improvement of grassland quality, and the relationship between the GECP and grassland quality. 

Some questions for consideration of the authors

1.      Briefly discuss the factors such as overgrazing, habitat loss, invasive species, climate change, and land management practices causing the vulnerability of grassland systems.

2.      What are the key indicators used in this study to assess the vulnerability of grassland ecosystems?

3.      If the grassland vulnerability changes have been studied before and after the implementation of the compensation policy.

4.      If stakeholders provided feedback and insights regarding the policy's effectiveness and relevance to addressing grassland vulnerability?

5.      How does the grassland ecological compensation policy compare to similar policies implemented in other regions or for grassland ecosystems in other countries?

6.      The DID model is a powerful tool, its validity depends on the underlying assumptions.  Please mention the assumptions and potential sources of bias when interpreting the results.

7.      What statistical software packages were used to perform moderation analysis?

Author Response

This is an interesting research analyzing the impact of grassland compensation policies on enhancing grassland quality by employing a county-level panel dataset from 2001 to 2021. A theoretical background provides the basis for defining the main hypothesis of the study. The vulnerability of the grassland ecosystem has been computed, and the association between GECP and grassland quality have been empirically tested using the difference-in-difference (DID), Dynamic effect model, and moderating effect models. The study has shown that the grassland ecological compensation policy had a positive impact on grassland quality and grassland ecosystem vulnerability.  The spatial differentiation pattern of grassland ecosystem vulnerability is well explained.  The positive impacts of policies become more pronounced as the vulnerability of grassland ecosystems decreases. The main conclusions from the study are well-articulated with emphasis on the spatial distribution of grassland ecosystem vulnerability, improvement of grassland quality, and the relationship between the GECP and grassland quality.

Point 1: Briefly discuss the factors such as overgrazing, habitat loss, invasive species, climate change, and land management practices causing the vulnerability of grassland systems.

Response 1: Thanks for the reviewer’s kind suggestion. We have made correction according to the reviewer’s Points. The revised details can be found in the red font section on lines 317-336.

Point 2: What are the key indicators used in this study to assess the vulnerability of grassland ecosystems?

Response 2: Thanks for the reviewer's kind suggestion. This paper first calculates the average annual grassland NPP of each county in the study area from 2001 to 2021, and uses this as the basic data to calculate the sensitivity and adaptability of the grassland ecosystem, and the vulnerability of the grassland ecosystem is equal to the difference between the sensitivity and the adaptability.The revised details can be found in the red font section on lines 185-225.

Point 3: If the grassland vulnerability changes have been studied before and after the implementation of the compensation policy.

Response 3: Thanks for the reviewer’s kind suggestion. Limited by the research method, the vulnerability of grassland ecosystems obtained in this paper is reflected based on the values of grassland NPP in all years from 2001 to 2021. Therefore, it is impossible to further study the changes in grassland ecological vulnerability before and after the implementation of the policy, which may be one of the directions for future research.

Point 4: If stakeholders provided feedback and insights regarding the policy's effectiveness and relevance to addressing grassland vulnerability?

Response 4: Thanks for the reviewer’s kind suggestion. At present, the government has formulated GECP in each province within the scope of the policy. the policy has been implemented depending on the size of the grassland area, excluding ecological factors such as the grassland super-class group and ecological function in most provinces and regions [1]. But the compensation standards of the SISGC are confusing and conflicting, and there has been no feedback relevance to addressing grassland vulnerability.

Point 5: How does the grassland ecological compensation policy compare to similar policies implemented in other regions or for grassland ecosystems in other countries?

Response 5: Thanks for the reviewer’s kind suggestion. Despite the importance of grasslands, however, few countries have created PES programs that protect their grasslands and the pastoralists whose livelihoods rely on them [2]. Therefore, this paper does not compare GECP with other grassland ecosystem policies.

Point 6: The DID model is a powerful tool, its validity depends on the underlying assumptions.  Please mention the assumptions and potential sources of bias when interpreting the results.

Response 6: Thanks for the reviewer’s kind suggestion. We have made correction according to the reviewer’s Points. The revised details can be found in the red font section on lines 383-386.

Point 7: What statistical software packages were used to perform moderation analysis?

Response 7: Thanks for the reviewer’s kind suggestion. This paper mainly uses STATA software and ArcGIS software to analyze the moderation effect.

References:

[1] Lin, Huilong, Yuting Zhao, and Ghulam Mujtaba Kalhoro. "Ecological response of the subsidy and incentive system for grassland conservation in China." Land 11.3 (2022): 358.

[2] Hou, Lingling, et al. "Grassland ecological compensation policy in China improves grassland quality and increases herders’ income." Nature Communications 12.1 (2021): 4683.

Reviewer 3 Report

Dear authors! The article is of high scientific and practical importance. Determining the methodology for evaluating the effectiveness of individual measures of state support for agriculture is a very important and necessary task, the solution of which is very important in terms of optimizing not only budgetary policy, but the entire economy as a whole.

The study leaves a favorable impression with the amount of data used and the scope of the task. The regulation of pasture quality in terms of economics and ecology is an important point in optimizing agricultural production.

Despite the positive aspects of the study, there are a number of comments that, in principle, do not reduce the value of the work done.

For example, in the introduction, the authors ask about the inconsistency of data from previous studies. Among all the assumptions made, for some reason, there is no question of the insufficient quality of the studies carried out. For example, some studies were based on data from 2011 (when the GECP program was launched) to 2014.

The second major problem of this work is an attempt to combine pastures with sharply different geographical conditions within the framework of the proposed model. This problem is clearly seen in Fig. 1 The classification of grassland ecosystem vulnerability. It is clearly shown that the vulnerability of pasture ecosystems depends not only on the factors indicated in the study, but also on their convenience, historical traditions and other factors.

Author Response

The article is of high scientific and practical importance. Determining the methodology for evaluating the effectiveness of individual measures of state support for agriculture is a very important and necessary task, the solution of which is very important in terms of optimizing not only budgetary policy, but the entire economy as a whole. The study leaves a favorable impression with the amount of data used and the scope of the task. The regulation of pasture quality in terms of economics and ecology is an important point in optimizing agricultural production. Despite the positive aspects of the study, there are a number of comments that, in principle, do not reduce the value of the work done.

Point 1: For example, in the introduction, the authors ask about the inconsistency of data from previous studies. Among all the assumptions made, for some reason, there is no question of the insufficient quality of the studies carried out. For example, some studies were based on data from 2011 (when the GECP program was launched) to 2014.

Response 1: Thanks for the reviewer’s kind suggestion. The current research on the ecological effects of GECP does have the problem of inconsistent results due to different data used. However, after combing through the literature, this article found that at the provincial scale, when comparing different provinces, the research results are quite different. But within a specific province, existing research results seem to be consistent. Therefore, this article combines the vulnerability of grassland ecosystems to analyze this possible regional heterogeneity, and does not deeply analyze the problem of data inconsistency in the assumptions made.

Point 2: The second major problem of this work is an attempt to combine pastures with sharply different geographical conditions within the framework of the proposed model. This problem is clearly seen in Fig. 1 The classification of grassland ecosystem vulnerability. It is clearly shown that the vulnerability of pasture ecosystems depends not only on the factors indicated in the study, but also on their convenience, historical traditions and other factors.

Response 2: Thanks for the reviewer’s kind suggestion. We have made correction according to the reviewer's Points. The revised details can be found in the red font section on lines 317-347. Considering that the study area is an area where ethnic minorities gather, and its historical and traditional culture is relatively diverse and complex, this article has not fully discussed it, and this will also be discussed in depth as a follow-up research direction.

Reviewer 4 Report

Please check The English Grammar.  

The manuscript is well written. It is a very good policy paper.  I have read the entire paper. The hypothesis is very good. It is my advise, please check the English Grammar. 

Author Response

Point 1: Please check The English Grammar.

Response 1: Thanks for the reviewer’s kind suggestion. We have made correction according to the reviewer’s Points. We have carefully revised the entire manuscript, including grammar, spelling, and sentence structure.
